# A Novel Symbiotic Beverage Based on Sea Buckthorn, Soy Milk and Inulin: Production, Characterization, Probiotic Viability, and Sensory Acceptance

**DOI:** 10.3390/microorganisms11030736

**Published:** 2023-03-13

**Authors:** Nicoleta-Maricica Maftei, Alina-Viorica Iancu, Roxana Elena Goroftei Bogdan, Tudor Vladimir Gurau, Ana Ramos-Villarroel, Ana-Maria Pelin

**Affiliations:** 1Faculty of Medicine and Pharmacy, Research Centre in the Medical-Pharmaceutical Field, University “Dunărea de Jos”, 800010 Galati, Romania; 2Medical Laboratory Department, Clinical Hospital of Children Hospital “Sf. Ioan”, 800494 Galati, Romania; 3Medical Laboratory Department, Clinical Hospital of Infectious Diseases “Sf. Cuvioasa Parascheva”, 800179 Galati, Romania; 4Biology and Animal Health Department, University of Oriente, Av. University, Campus the Guaritos, Maturín 6201, Venezuela

**Keywords:** symbiotic, health, probiotic, inulin, beverage

## Abstract

Nowadays, vegan consumers demand that food products have more and more properties that contribute to the prevention of some diseases, such as lower fat content, increased mineral content (calcium, iron, magnesium, and phosphorus), pleasant flavor, and low calorie values. Therefore, the beverage industry has tried to offer consumers products that include probiotics, prebiotics, or symbiotics with improved flavor and appearance and beneficial effects on health. The possibility of producing beverages based on soy milk with sea buckthorn syrup or sea buckthorn powder supplemented with inulin and fermented with the *Lactobacillus casei* ssp. *paracasei* strain was examined. The aim of this study was to obtain a novel symbiotic product that exploits the bioactive potential of sea buckthorn fruits. Tests were carried out in the laboratory phase by fermenting soy milk, to which was added sea buckthorn syrup (20%) or sea buckthorn powder (3%) and inulin in proportions of 1% and 3%, with temperature variation of fermentation (30 and 37 °C). During the fermentation period, the survivability of prebiotic bacteria, pH, and titratable acidity were measured. The storage time of beverages at 4 °C ± 1 °C was 14 days, and the probiotic viability, pH, titratable acidity, and water holding capacity were determined. Novel symbiotic beverages based on sea buckthorn syrup or powder, inulin, and soy milk were successfully obtained using the *Lactobacillus casei* ssp. *paracasei* strain as a starter culture. Moreover, the inulin added to the novel symbiotic beverage offered microbiological safety and excellent sensory attributes as well.

## 1. Introduction

Today, all around the world, people’s desire to consume functional drinks is huge, and for this reason, they have begun to be added to the modern medicine cabinet.

From a commercial and medical point of view, there is a wide variety of functional beverages that can meet the different nutritional requirements and ingredient preferences of consumers around the world. Some of the common categories include vegetable and fruit-based drinks, whey and soy protein drinks, sports drinks, and tea-based drinks. A few examples can be mentioned, such as kvass, produced from rye flour or stale rye bread in several Eastern Europe countries and Russia [1]; borş, a liquid obtained by natural fermentation of an aqueous suspension of wheat bran—“tărțe” means “bran” in Romanian—and corn flour [2], green mung beans, and red kidney beans [3]; fruit beer; water kefir; vinegar [4]; boza, obtained from millet, corn, wheat, or rice and consumed in Bulgaria, Albania, Turkey, Greece, and Bosnia Herzegovina [5]; Jerusalem artichoke [6]; prickly pear juice [7]; and apricot juice [8].

Among all functional food categories, the functional beverage market is the fastest growing, and by 2025, it is expected to increase to 40% of entire consumption [9].

Sea buckthorn (*Hippophae* L.) is a valuable plant extensively characterized by a unique composition of bioactive compounds. Its use is growing in Europe, Canada, Asia, and the USA [10]. Due to high amounts of antioxidants, it is widely used for nutraceutical and medicinal purposes. Apart from its antioxidant capacity, sea buckthorn was observed to have antibacterial and antiviral [11], hepatoprotective and dermatological [12], antidiabetic [13], anti-inflammatory [14], and anticarcinogenic [15] effects. The positive biological, physiological, and medicinal effects of sea buckthorn were extensively described in a multitude of studies [16,17,18,19,20].

Vegetable beverages are extracts from legumes, plants, cereals, seeds, etc., in water and have been used worldwide as a replacement for cow’s milk. There is a large market for vegetable beverages all around the world, but soy drinks are the most consumed [21]. According to a report by Grand View Research, the soy beverage segment led the industry in 2021 and accounted for the maximum share of more than 38.40% of the overall revenue [22]. The global soymilk market size was valued at USD 7.30 billion in 2018 in a market analysis report [23]. The main reason for the huge consumption of vegetable and plant-based drinks such as soybean beverages, to the detriment of dairy products, is caused by growing consumption by people with medical problems such as allergy to cow’s milk proteins, lactose intolerance, and/or cholesterol problems [24]. In recent years, globally, consumers have accepted the use of vegetable and plant sources to produce functional foods due to their bioactive components that present better health benefits, which also favored the increase in their consumption [25].

Inulin is a natural polysaccharide that belongs to a class of dietary fibers that are usually found under the name of fructans [26]. The compound that results from the combination of inulin with fructooligosaccharides can be used as a dietary non-digestible fiber. It is often used in food industries because it promotes a beneficial effect on the host’s gut microbiota by stimulating the growth of bifidobacteria in the human intestine. The main reason inulin is used is its prebiotic effect. On the other hand, it is also utilized for the control of sugar levels in the blood, prevention of obesity, lipid metabolism, the absorption of mineral ions from the gut, and colon cancer [27]. These are some extra benefits of inulin regarding the beneficial effects on the human body.

Because the fermentation process involves mixed cultures of LAB, or fungi and yeast, traditional fermented foods are an abounding source of microorganisms, and some show probiotic characteristics [28]. In the literature, it is mentioned that upon ingestion of probiotics (viable microorganisms) in adequate amounts, health benefits are exerted, which improve the intestinal microbial balance. Lactobacilli and bifidobacteria are the most used probiotic that can survive in the intestinal tract. Moreover, by combining probiotics and prebiotics (i.e., symbiotics), synergistic benefits may be observed. However, there is a small number of studies on the development of symbiotics, and the in vivo and in vitro effectiveness of symbiotics has not been intensively studied to date. Innovations and novel product development come with their own set of challenges, and each matrix (vegetable or fruit) is unique. These products require the standardization of the process to obtain a product with acceptable organoleptic properties and, more importantly, demonstrable health benefits. Hence, this study aimed to develop a novel symbiotic beverage based on lactic acid bacteria, soy milk, and sea buckthorn syrup or powder enriched with inulin to evaluate the physicochemical, nutritional, bioactive, and sensory characteristics of the formulation obtained. The viability of probiotic bacteria, the major physicochemical properties during 6 h of fermentation and 14 days of storage at 4 °C and the sensory properties of these products were analyzed.

## 2. Materials and Methods

### 2.1. Materials

#### 2.1.1. Soy Milk

In this experiment, Dr. Oetker soymilk–Soy Beverage Inedit from Company, Romania—was used. This is a sterilized product obtained from selected organic-certified ingredients. Per the labeled data, the product contains the following ingredients: 1.1% proteins, 0.0% sugars, and 1.9% lipids.

#### 2.1.2. Sea Buckthorn Syrup

The sea buckthorn syrup used in these experiments came from the Plafar market in Galati, Romania. The untreated syrup contains the following soluble solids: 6° Brix with a pH of 3.10. All samples utilized during the experiment came from the same batch.

#### 2.1.3. Plant Material for Sea Buckthorn Powder

Sea buckthorn berries were collected from the region of Moldavia (Romania) in September 2020 and were stored at −20 °C prior to the experiments. Sea buckthorn fruits were previously subjected to washing, sorting, and cleaning operations and were squeezed with the help of a mixer, and the collected juice was freeze-dried with a Freeze-dryer Alpha 1-4 LDplus (Martin Christ Gefriertrocknungsanlagen GmbH, Osterode am Har, Germany).

#### 2.1.4. Starter Cultures (Probiotic Bacteria)

*Lactobacillus casei* ssp. *paracasei* was provided by Christian Hansen (Hørsholm, Denmark) as a starter culture with the commercial name *L. casei*^®^ 431 in a freeze-dried form. Starter culture maintenance and storage were carried out per the manufacturer’s recommendation.

### 2.2. Fermentation of Beverages and Analytical Assays

Four pilot-scale beverage-making trials with:(a)20% sea buckthorn syrup (*w*/*w*) in milk and 1% (*w*/*v*) commercial inulin;(b)20% sea buckthorn syrup (*w*/*w*) in milk and 3% (*w*/*v*) commercial inulin;(c)3% (*w*/*v*) sea buckthorn powder in milk and 1% (*w*/*v*) commercial inulin;(d)3% (*w*/*v*) sea buckthorn powder in milk and 3% (*w*/*v*) commercial inulin, respectively, were prepared. A total quantity of 1500 g per beverage formulation was produced per each trial, and the whole experiment was repeated three times.

After mixing the soy milk with sea buckthorn syrup or sea buckthorn powder and inulin, the mixtures were homogenized separately with an Ultra Turrax blender (IKA, Merck, Darmstadt, Germany) at 14,000 rpm until all ingredients were dissolved in the soy milk. Then, 100 mL mixtures were inoculated with starter culture and were fermented for 6 h at 30 and 37 °C. Samples were taken during the incubation at 0, 2, 4, and 6 h to test the following parameters: starter culture growth, pH, and titratable acidity.

After fermentation, the samples were stored for 14 days at 4 °C ± 1 °C. During the entire storage time, the viability of probiotic bacteria, pH, titratable acidity, and water-holding capacity were measured.

A pH meter (MP2000, Mettler Toledo, Greifensee, Switzerland) was used to measure the pH. A solution of 0.1 N NaOH was used to determine the titratable acidity, and it was expressed in grams of lactic acid per 100 mL of fermented product.

Immediately after preparation, its physico-chemical characterization was determined. An Atago RX-1000 refractometer (Atago Company Ltd., Tokyo, Japan) was used for the soluble solid content, and an InoLab Multilevel 1 conductive meter (Senton, GmBh, Egelsbach, Germany) was used for the electrical conductivity of the beverages.

### 2.3. Probiotic Viability

DeMan–Rogosa–Sharpe (MRS) agar was used to grow lactic acid bacteria (*Lactobacillus casei* ssp. *paracasei*). A series of dilutions using 0.1% (*w*/*v*) peptone water (Merck) were prepared by aseptically removing the diluent of culture until the dilution factor determined in the preliminary test was achieved. Using the spread plate method, 0.1 mL of the sample was transferred onto the agar, and plates were incubated at 37 °C for 48 hrs. Anaerobic conditions were established using an anaerobic jar (Merck) with Anaerocult^®^ A kit (Merck). Plates containing from 30 to 300 colonies were chosen for enumeration, which was expressed in colony-forming units per milliliter of product (CFU·mL^−1^). Analyses were performed in triplicate.

### 2.4. Water-Holding Capacity (WHC)

To measure the water-holding capacity (WHC) in the beverages, approximately 10 g of beverage was transferred into a 20 mL glass tube and centrifuged (Mikro 220R, Hettich, Tuttlingen, Germany) at 2500 rpm for 10 min at 20 °C (modified method of [29]). The WHC was calculated as follows: Water-holding capacity,
% = (weight of supernatant/weight of beverage) × 100(1)

### 2.5. Rheological Measurements

For the study of the beverage’s rheological properties, a rheology assay was carried out immediately after the manufacturing of each beverage formulation using an RHEOTEST-2 type rotating viscometer manufactured by VEB-MEDINGEN, Germany. All samples were gently stirred before rheological analysis. A total of 50 g of the sample was tested, and the coaxial cylinder device S3 was used due to the medium viscosity of the samples. The shear rate (*γ*) varied between 0.1667 to 145.6 s^−1^, and the working frequency was 50 Hz. The apparent viscosity (η) was calculated as:(2)$$\eta =\frac{\tau }{\gamma }$$
where *γ* = shear rate and η = apparent viscosity.

### 2.6. Sensory Evaluation

Sensory evaluation was carried out after 24 h storage at 4 °C. The hedonic scale was used to determine the degree of sensory acceptance for symbiotic beverages based on soy milk, sea buckthorn syrup or powder, and inulin. Ten untrained panelists were given 10.0 g of each sample. The samples were served at 8 ± 2 °C in plastic cups labeled with three-digit codes and were evaluated for aroma, texture, taste, and overall acceptability based on a seven-point hedonic scale (1 = dislike extremely, 5 = either like or dislike, 7 = like extremely).

### 2.7. Statistical Analysis

All the experiments were carried out in triplicate, and the data were analyzed using the Statgraphics plus v.5.1 package (Manugistics Inc., Rockville, MA, USA) with one-way analysis of variance. The results were presented as mean ± S.D (standard deviation). The comparison of means was conducted using Duncan’s Multiple Range Test (DMRT) with values that have no common superscript significantly different (*p* < 0.05).

## 3. Results

### 3.1. Microbial Growth and Physico-Chemical Parameters during the Fermentation Period

The counts for *Lactobacillus casei* ssp. *paracasei* (*L. casei*^®^ 431) strain in fermented beverages are shown in Figure 1a,b. In all samples, *L. casei*^®^ 431 grew without the need to add nutrient supplementation. After 6 h of fermentation, the microbial load (*L. casei*^®^ 431 strain) increased exponentially in the product to around 1.13·10^8^ CFU·mL^−1^ for sea buckthorn powder (SBP) beverages and 1.11·10^8^ CFU·mL^−1^ for sea buckthorn syrup (SBS) beverages (Figure 1a,b), starting from an initial concentration of 4.8·10^5^ CFU·mL^−1^ from the sample with SBS and 5.2·10^5^ CFU·mL^−1^ from the sample with SBP. The multiplication rate increased progressively for all samples (*p* < 0.05). For the sample with 3% inulin and SBP (5.2·10^7^ CFU·mL^−1^) fermented at 37 °C, significantly more growth was observed.

The pH values of all beverages (Figure 2a,b) decreased from the initial pH of 4.21, 4.01 to 3.87 and 3.88 for the SBS samples fermented at 30 °C and 37 °C with 1% and 3% inulin, respectively. For SBP beverages, the pH values after 6 h of fermentation were similar—3.89 at 30 °C for both beverages and 3.86, 3.92 at 37 °C for beverages with 1% and 3% inulin, respectively. The pH decrease was caused by the production of organic acids.

As shown in Figure 3a,b, the titratable acidity of the samples increased alongside the decrease in pH for all samples. The titratable acidity reached 1.07 g lactic acid 100 mL^−1^ for the sample with 1% inulin and 0.89 g lactic acid 100 mL^−1^ for the sample with 3% inulin and SBS fermented at 30 °C, post 6 h of fermentation. For the SBS samples fermented at 37 °C, the titratable acidity reached 1.08 g lactic acid 100 mL^−1^ and 0.93 g lactic acid 100 mL^−1^. At the end of the fermentation, there was approximately 0.65 g lactic acid 100 mL^−1^ and 0.48–0.74 g lactic acid 100 mL^−1^ for SBP beverages with 1% and 3% inulin, respectively, fermented at 30 °C and 37 °C.

The physicochemical and rheological characteristics are presented in Table 1 and Table 2 for all beverages before inoculation with probiotic bacteria. The increase in the soluble solids content and the decrease in pH were caused by the increase in inulin content. The viscosity of the beverages increased with the concentration of inulin (Table 2), regardless of the fermentation temperature. For SBS beverages, the increase in inulin concentration percent decreased the acidity, as observed in Table 1. For SBP beverages, percent acidity increased with the increase in inulin concentration. The electrical conductivity was the same for beverages with 3% inulin and increased for beverages with 1% inulin. During the fermentation period, the increase in acid production, as demonstrated by the lower pH and higher titratable acidity values, was associated with the consumption of TSS. In conclusion, these are indicative of the post-acidification microbiological process.

### 3.2. Survivability of Probiotics and Physicochemical Parameters during the Storage Period

It can be observed from our results that the *L. casei*^®^ 431 strain has good survival rates without nutrient supplementation in all beverages. Figure 4a,b presents the changes in the cell viability of the *L. casei*^®^ 431 strains (*p* < 0.05) for the 14 days of cold storage at 4 ± 1 °C. This evaluation began after the end of the 6 h of fermentation. There was a reduction in the viability of *L. casei*^®^ 431 cells after one day of storage in all formulations. At the end of the storage period, the total viable cells of *L. casei*^®^ 431 were decreased moderately to 1.11·10^7^ for the SBS beverage with 1% and 1.15·10^7^ for the SBS beverage with 3% inulin, fermented at 30 °C, and 1.14·10^7^ and 1.15·10^7^ CFU·mL^−1^ for the SBS beverage with 1% and 3% inulin, fermented at 37 °C (Figure 4a), respectively. However, the SBP beverage with 1% and 3% inulin fermented at 30 °C and 37 °C presented a bigger population of *L. casei*^®^ 431 compared to all the formulations (*p* < 0.05). After the storage period, the viability of *L. casei*^®^ 431 remained at an acceptable level (over 10^7^ CFU·mL^−1^).

Figure 5a,b presents the pH evolution (*p* < 0.05) for all samples during the storage period. The pH ranges between 3.61 and 3.64 for the SBS beverage with 1% inulin fermented at 30 °C and 37 °C, respectively, and 3.59 for SBS with 3% inulin for both temperatures of fermentation after storage time. However, for the SBP beverages, the pH decreased gradually over time (*p* < 0.05), ranging between 3.57–3.64 and 3.54–3.67 for samples fermented at 30 °C and 37 °C, respectively.

One of the metabolites associated with functional and nutritive characteristics of fermented beverages—lactic acid—was determined. In Figure 6a,b (*p* < 0.05), the values for all beverages after the storage time are shown.

After 14 days of refrigerated storage, the acidity values increased (*p* < 0.05) in all beverages, except for the SBP beverage with 1% inulin fermented at 37 °C, which showed a lower increase in this parameter in the time of storage.

During the period of storage, the WHC was higher for the beverages fermented at 30 °C and 37 °C (Figure 7a,b). Regardless of the inulin, the WHC increased with the increase in inulin concentration. As shown in Figure 7, the WHC of SBS and SBP fermented at 30 °C and 37 °C, respectively, with 3% inulin, was significantly higher than that of other groups (*p* < 0.05). SBP beverage with 3% inulin fermented at 37 °C had the largest water-retention capacity, while the WHC of SBS was only 71.96%, which was the weakest.

### 3.3. Sensory Evaluation

The sensory analysis results are shown in Table 3. Significant differences were observed between all beverages (*p* < 0.05). The SBS and SBP samples with the highest content of inulin (3%) fermented at 37 °C had a more intense color and were, therefore, better scored by the panelists. Regardless of the attributes of flavor and taste, there are no significant differences (*p* > 0.05) within samples (fermented at both temperatures) supplemented with inulin. Formulation SBP (3% inulin, fermented at 37 °C) (Table 3) had the highest values for flavor, taste, and overall acceptability attributes, differing significantly from the others (*p* < 0.05).

## 4. Discussion

The beneficial effects of prebiotics and probiotics in diets have been and still are under exploration. Results provided from a limited number of contradictory studies create difficulties in developing a general recommendation for obtaining a novel functional beverage. Nazir et al. [30] declared that science-based demonstrations are necessary to validate health claims and successfully market new functional beverages.

This study examined the possibility of producing a symbiotic beverage based on soy milk with sea buckthorn syrup or sea buckthorn powder supplemented with inulin and fermented with the *Lactobacillus casei* ssp. *paracasei* strain.

Studies that have examined the effects of multiple prebiotics on food consumption, such as inulin, oligosaccharides, fructooligosaccharides, and soy milk, sea buckthorn syrup, or sea buckthorn powder fermented with lactic acid bacteria, showed inconsistent results.

From what we know, it appears that there are no studies available reporting on the symbiotic beverages based on soymilk, sea buckthorn syrup, or sea buckthorn powder and inulin; in this regard, it proves to be a difficult task to compare our results with those reported for other beverages because of the differences in the used experimental conditions, food matrix, studied microorganisms, and the used analytical and instrumental methods and apparatus.

The cell viability of *L. casei^®^* 431 was not affected by either the time, concentration, or temperature. However, at the 6th h of fermentation, its number increased, and the results indicated that there was competition for the nutrients. The increase in cell numbers was higher at 37 °C for the SBP beverage with 3% inulin, compared to 30 °C and 37 °C, for the SBS beverage with 1 and 3% inulin. As shown in Figure 1a,b supplementation with different amounts of inulin in the beverage affects the growth of *L. casei ^®^* 431; the cell numbers in the samples increased with the amount of inulin.

During storage, the cell numbers of *L. casei ^®^* 431 slightly increased and remained at 1.4·10^8^ and 1.9·10^8^ CFU mL^−1^ for the SBP beverage with 3% inulin fermented at 30 °C and 37 °C, respectively (Figure 4a,b). Generally, during storage, cell numbers of *L. casei ^®^* 431 increased slightly for all beverages at both temperatures of fermentation. Analysis of variance for the probiotic counts showed that supplementation with different amounts of inulin in the beverage affected the growth of *L. casei ^®^* 431; the cell numbers in the samples increased with the amount of inulin both during fermentation and the storage period. The recommended level of viable probiotic cells at the time of food consumption is 10^7^–10^8^ CFU·mL^−1^; after 14 days of storage at 4 °C [31], our viable cells of *L. casei ^®^* 431 remained at the level mentioned. No less than a million viable cells/mL of the probiotic product are needed to reach the minimum amount of health benefits for consumers, as [32] mentions. Our results demonstrate that the *L. casei ^®^* 431 culture strain can be used as a probiotic for obtaining symbiotic beverages based on soymilk, sea buckthorn syrup, or sea buckthorn powder and inulin. In conclusion, we can declare that it is very important to test the compatibility between probiotics and prebiotics to provide a positive interaction capable of increasing microbial viability during the fermentation and storage period. We can also confirm the use of good preparation practices because no microbial contaminants were detected in the symbiotic beverages during the storage period.

The pH values of all samples decreased from the initial pH after 6 h of fermentation and showed nearly the same final pH. The decrease in pH was possibly due to the accumulation of lactic acid. It should be mentioned that fermented products based on soy milk are dependent on pH because it influences the stability, aroma, flavor, and texture of the final product. The product becomes too acidic, and the soybean proteins precipitate when pH < 4.0. Behrens et al. [33] declared that at a pH above 4.5, the flavor of the product is conserved. The improvement in the sensory quality of soymilk is obtained by the masking of volatile soybean compounds (n-hexanal and pentanal) by the fermentation products, especially compounds such as lactic acid, diacetyl, and acetaldehyde [33].

Likewise, for symbiotic beverage production, it is very important to have enhanced acid production during fermentation because it depends upon the growth, viability, and ability of *L. casei* to ferment the milk carbohydrates [34]. In conclusion, the pH of the beverages must be stabilized at higher values.

The reduction in pH may be due to the release of organic acids following the post-acidification phenomenon, a phenomenon attributed to the fermentation of probiotics during refrigeration. The post-acidification phenomenon, thus, affects the viability of the starter culture [35]. However, Filannino et al. [36] suggest that the main advantage of the decrease in pH due to the accumulation of lactic acid during the fermentation period is better shelf-life preservation of the food. Additionally, the absence of coliform and mold counts during microbiological quality control may be correlated with the phenomenon of accumulation of lactic acid during the fermentation period.

Our results agree with the results reported by Narvhus et al. [37]. They used lactococcal fermentation to obtain fermented milk at 22, 30, and 37 °C. After the fermentation, they observed that the products incubated at a temperature of 37 °C had a faster decrease in pH at the beginning of the fermentation period, but at the end of fermentation, the products incubated at 30 °C, as well as those incubated at 37 °C, had the same final pH [37].

The titratable acidity increased with the decrease in pH for all symbiotic beverages during fermentation for both temperatures. The SBS beverages had a higher value of titratable acidity compared with SBP during the fermentation and storage period. These differences could be assigned to the presence of 3% more solids of nonlipid origin (sea buckthorn powder) in the SBP beverages. Additionally, the titratable acidity increased with the decrease in pH during fermentation for both temperatures and for all beverages. Likewise, Gardner et al. [38] reported using mono and mixed cultures of lactic bacteria to obtain fermented vegetable juice. The amount of lactic acid varied in the range of 0.3 g·mL^−1^ to 1.5 g·mL^−1^ [24]. Guzel–Seydim et al. [39] mentioned that the acidity of fermented beverages is commonly maintained or decreased during the storage period. This fact is attributed to microbial proteolysis. Costa dos Santos et al. [40] stated that the phenomenon of microbial proteolysis is a continuous fermentation process during which the lactic acid bacteria assimilate the lactate present in the environment.

Electrical conductivity is an important parameter for evaluating the quality of any juice because it gives an idea of the freshness of the product. Moreover, electrical conductivity is influenced by the pH of the solution, the valence of the ions, and the degree of ionization. The values obtained in our study for electrical conductivity might be related to the components or vitamins released from sea buckthorn or inulin. Similar remarks were made by Boubezari et al. [41] for a probiotic juice based on carrots and *L. plantarum J12.*

The sea buckthorn powder significantly influenced the rheological parameters of the symbiotic beverage, which may be related to changes in the microstructure (e.g., inulin crystallization, gel formation) caused by the addition of inulin. The consistency index (k) was influenced by different formulations. However, the addition of inulin increased this index, and the association of inulin and SBP influenced the consistency index (*p* < 0.05). These results suggest that the combination of inulin with SBP had a synergistic effect, changing the beverage’s consistency. Similar studies evaluating the effect of inulin on the rheological parameters were reported by [42], which declared that viscosity was influenced by inulin. Additionally, observed differences in viscosity between samples with inulin and samples without were mentioned in [43].

The WHC values obtained on the first day of sampling were lower than those found during storage and significantly increased with increasing inulin concentrations. Our results agree with the results reported in [44]. In contrast, Lin et al. [45] reported that during the storage period, for beverages based on *L. chinense Miller* juice, milk and soy milk were fermented with mixed cultures composed of the following strains: *Bifidobacterium longum* and *Lactobacillus paracasei* subsp. *paracasei NTU101*, no syneresis phenomenon was observed.

Our results show that with the increase in inulin concentration percent, the viability of probiotic culture and acidity increased, while the water holding capacity and pH decreased for all symbiotic beverages.

Regarding sensory analysis, due to the expanding influence of cultures from all over the world, Terpou et al. [46] declared that today, most consumers are much more open to tasting new flavor combinations, especially with ingredients that can confer beneficial effects in terms of health.

The content of sea buckthorn powder and the presence of a probiotic and prebiotic in SBP samples could be the reasons for their greater acceptability than SBS beverages. Rinaldoni et al. [47] mentioned that ingredients such as inulin and protein concentrates are often used in the food and beverage industry to prepare food products, beverages, and desserts based on soy milk, not only to replace fats but also to provide special properties to the products (e.g., functional and nutritional properties). In this regard, the simultaneous addition of *L. casei ^®^* 431 and inulin improved the texture, color, and flavor of the symbiotic beverages. Considering the overall results of the sensory analysis, one can declare that the addition of inulin in beverages based on soy milk and sea buckthorn syrup or powder fermented with probiotic bacteria can improve the degree of acceptability of these symbiotic drinks. However, much more future study is needed that focuses on improving the taste and sensory acceptability of these types of beverages. The results obtained in the sensory evaluation indicate that inulin can improve the sensory attributes of the symbiotic beverage. We can also declare that inulin, together with soy milk, sea buckthorn, and probiotic culture, make our product a healthy beverage.

Iraporda et al. [48] observed a satisfactory growth of *L. paracasei* BGP1 in the soy-based formula. Moreover, they reported that fermented soy-based beverages with inulin represent a good matrix for probiotic delivery. Fermented soy-based beverages with inulin are an alternative to traditional dairy probiotic beverages and contribute to diversifying this type of agri-based food products [48]. Research groups worldwide continue to explore fruit and vegetable juice matrices for new probiotic product development. For instance, [49] incorporated initial populations of *L. acidophilus*, *L. rhamnosus*, *L. paracasei* ssp. *paracasei*, *Sacch. Boulardii*, and *B. lactis* into a non-fermented vegetarian frozen soy dessert made with a soy beverage, sugar, oil, stabilizer, and salt [50] using a mix of two starters (*S. thermophilus* and *L. delbrueckii* subsp. *bulgaricus*) to inoculate either cows’ milk or soy beverage with either the probiotic bacteria *L. rhamnosus*, *L. johnsonii*, or human-derived bifidobacteria. Regardless of the substrate used, a major problem with using probiotics in fruit juices is their viability in an acidic medium, which could be solved by proliferation-promoting compounds such as inulin, which is projected in the context of symbiotic products [51].

From this point of view, it was a difficult task to compare our results with those reported for other beverages because of the differences in the experimental conditions, food matrix, and the microorganisms studied by us and the ones mentioned in the literature.

## 5. Conclusions

Inulin was successfully incorporated as a prebiotic to produce a novel symbiotic beverage based on soymilk, sea buckthorn syrup or sea buckthorn powder, and inulin. SBP beverages had high survival rates during the 14 days of storage compared to SBS beverages. As a matter of fact, the achieved viabilities of the probiotic culture constantly surpassed the levels of the threshold (10^6^–10^7^ CFU·mL^−1^) that are needed to confer health benefits during the time of consumption of a probiotic beverage.

Following the experiments, it can be concluded that inulin can be used to obtain a symbiotic product based on soy milk and sea buckthorn juice/powder fermented with *L. casei ^®^* 431, but the pH of the product must be stabilized at higher values. Moreover, the inulin added to the novel symbiotic beverage offered microbiological safety and excellent sensory attributes. Sea buckthorn powder and inulin in the symbiotic beverage not only satisfied the role of a functional ingredient, but the symbiotic beverages produced had a better mouthfeel. Finally, because the produced symbiotic beverages combine the health benefits of both prebiotic and probiotic characteristics, they show a high commercialization potential within the beverage industry. As a matter of fact, this inulin-based beverage is an alternative to either dairy or vegetable–soy beverages and is suited to a healthy lifestyle, whether a vegetarian diet or not.

## Figures and Tables

**Figure 1 microorganisms-11-00736-f001:**
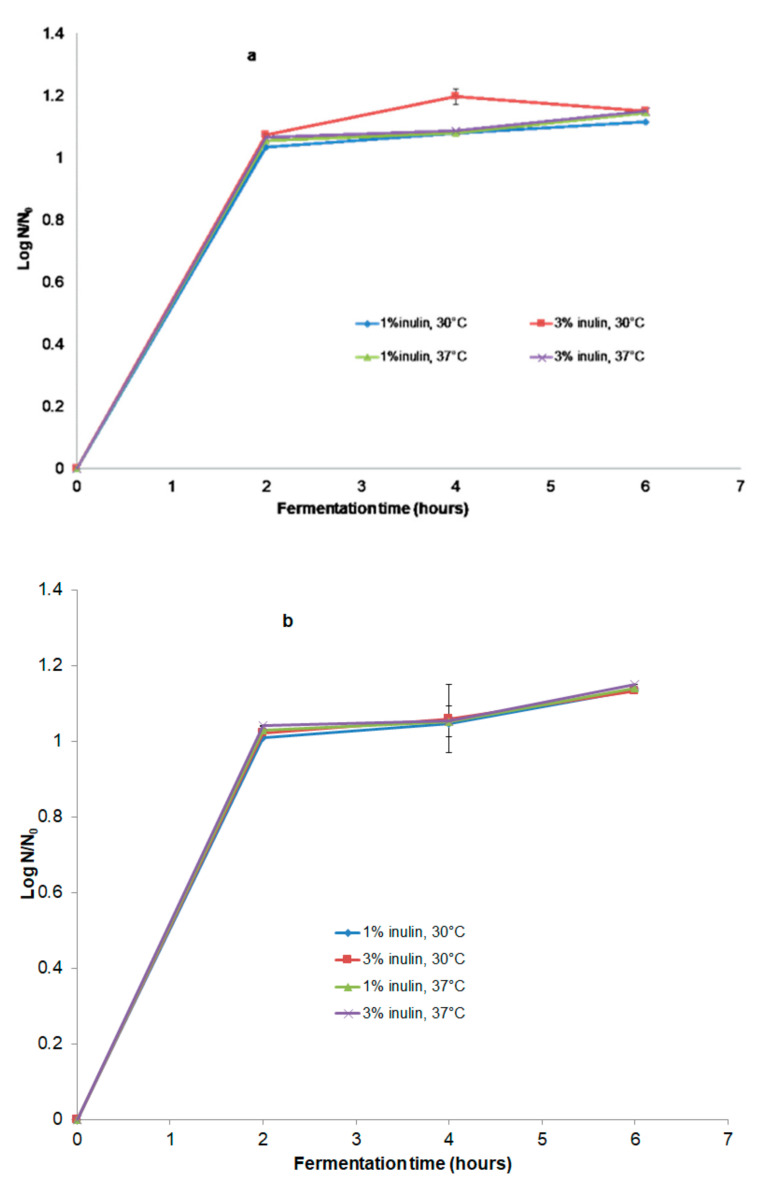
Probiotic viability during fermentation. (**a**) Samples with SBS; (**b**) Samples with SBP.

**Figure 2 microorganisms-11-00736-f002:**
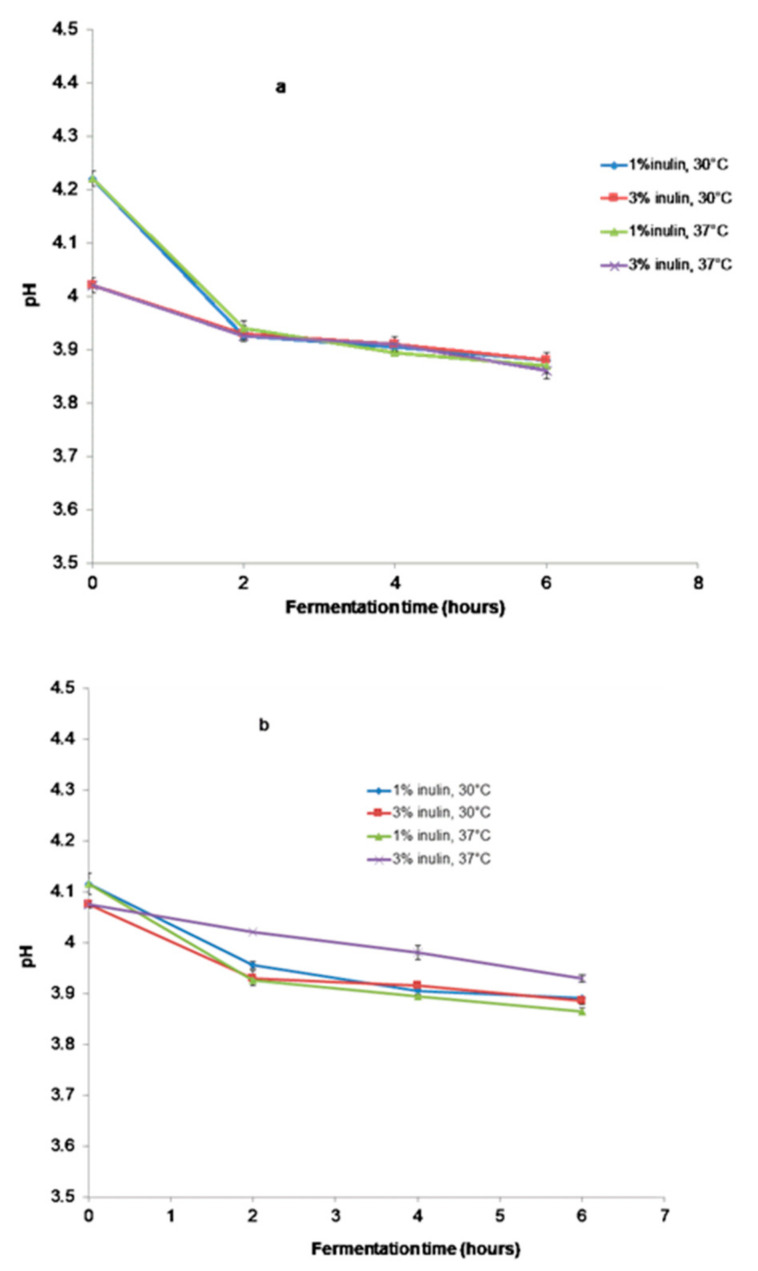
pH changes in beverages during fermentation (**a**) Samples with SBS; (**b**) Samples with SBP.

**Figure 3 microorganisms-11-00736-f003:**
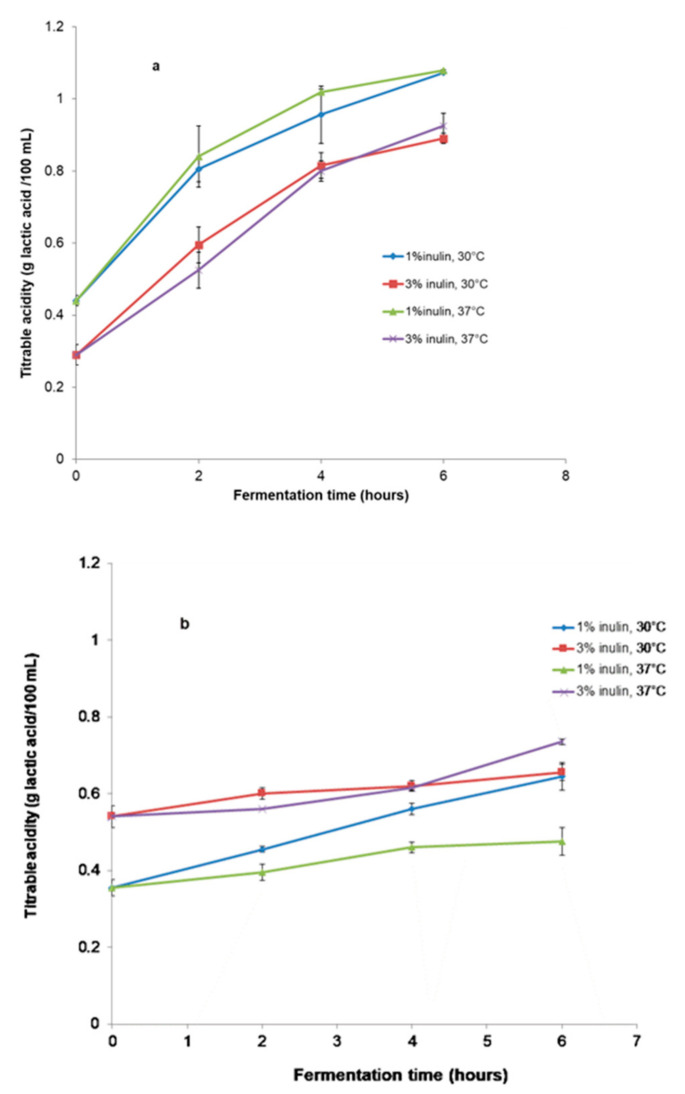
Variations in titratable acidity of beverages during fermentation. (**a**) Samples with SBS; (**b**) Samples with SBP.

**Figure 4 microorganisms-11-00736-f004:**
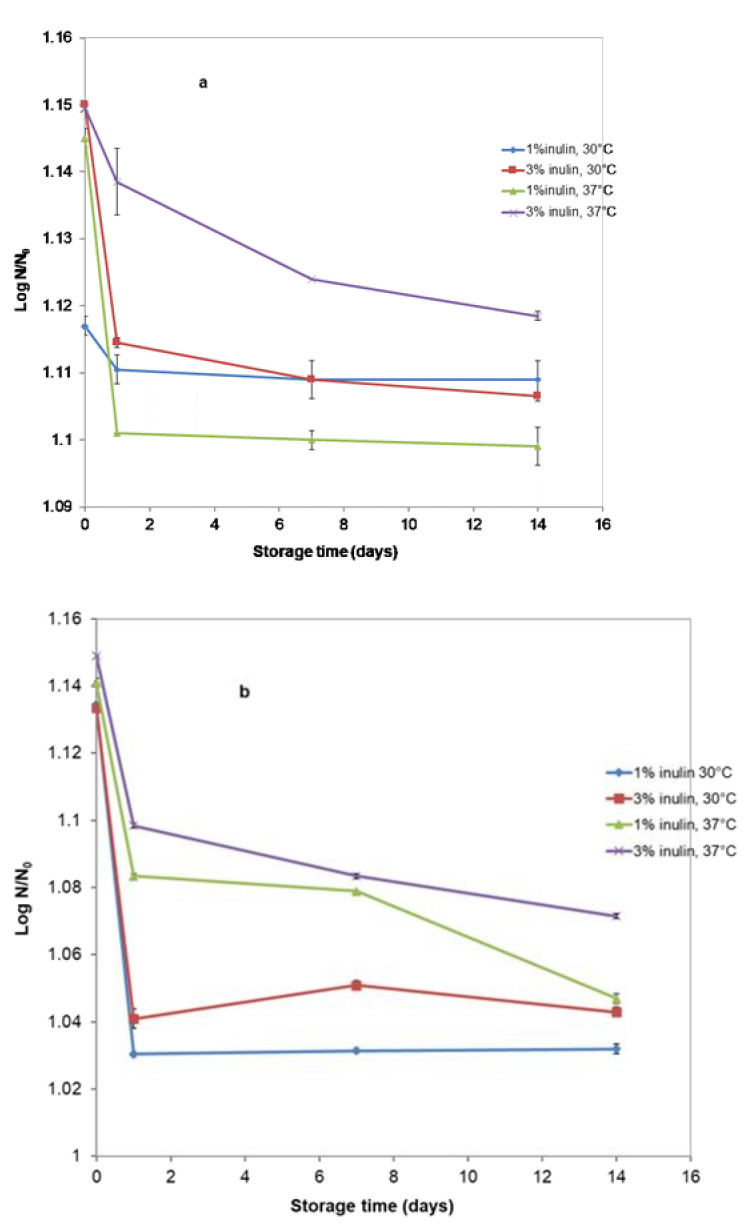
Viability of probiotic bacteria in the stored samples. (**a**) Samples with SBS; (**b**) Samples with SBP.

**Figure 5 microorganisms-11-00736-f005:**
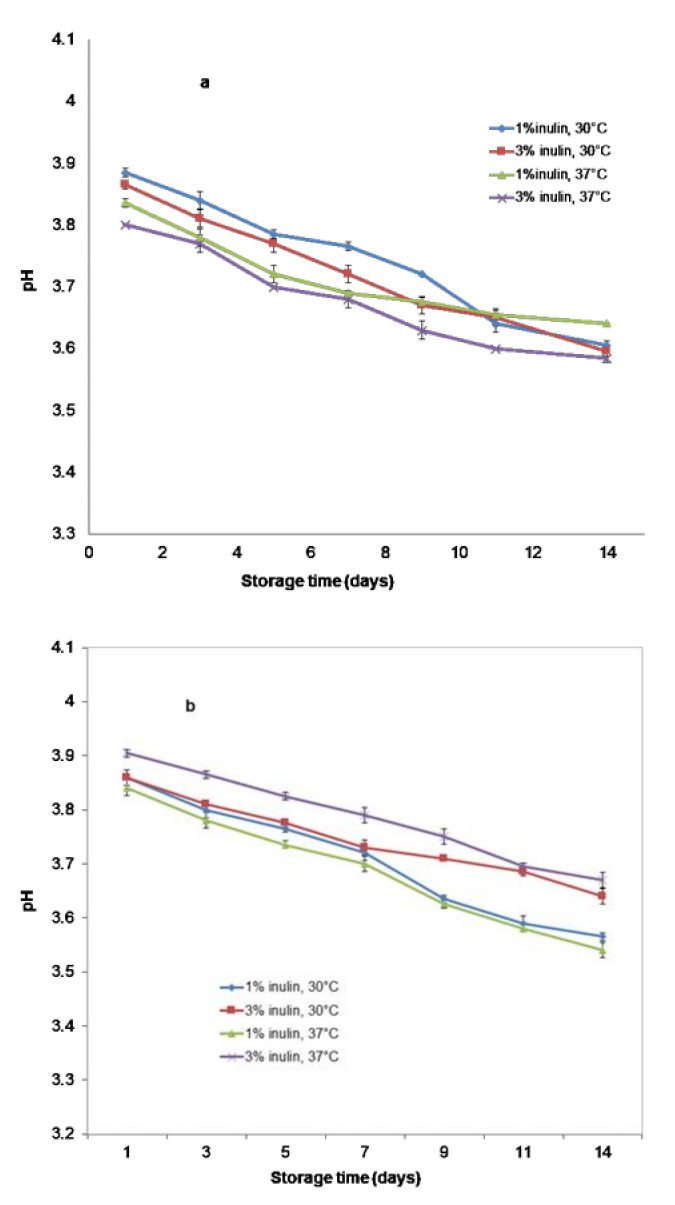
pH changes in beverages during storage. (**a**) Samples with SBS; (**b**) Samples with SBP.

**Figure 6 microorganisms-11-00736-f006:**
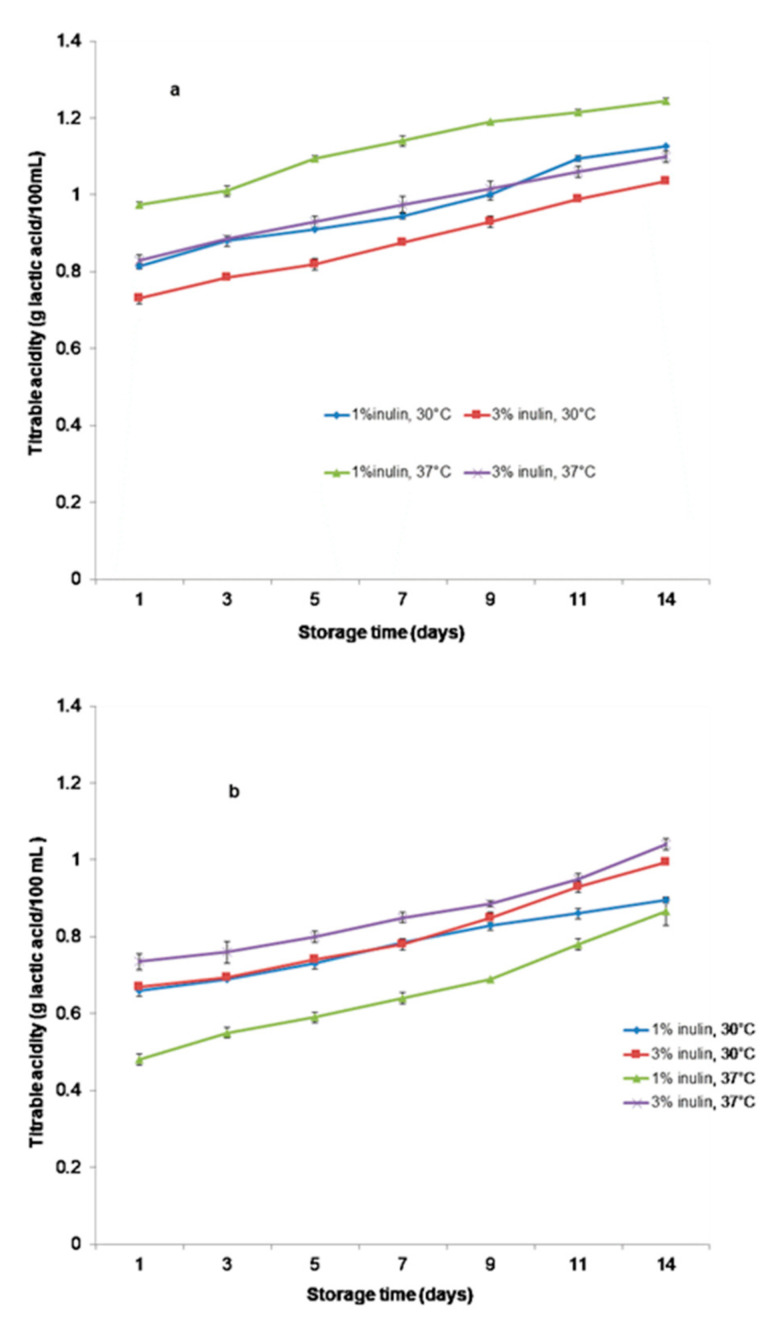
Variations in titratable acidity of beverages during storage at 4 °C. (**a**) Samples with SBS; (**b**) Samples with SBP.

**Figure 7 microorganisms-11-00736-f007:**
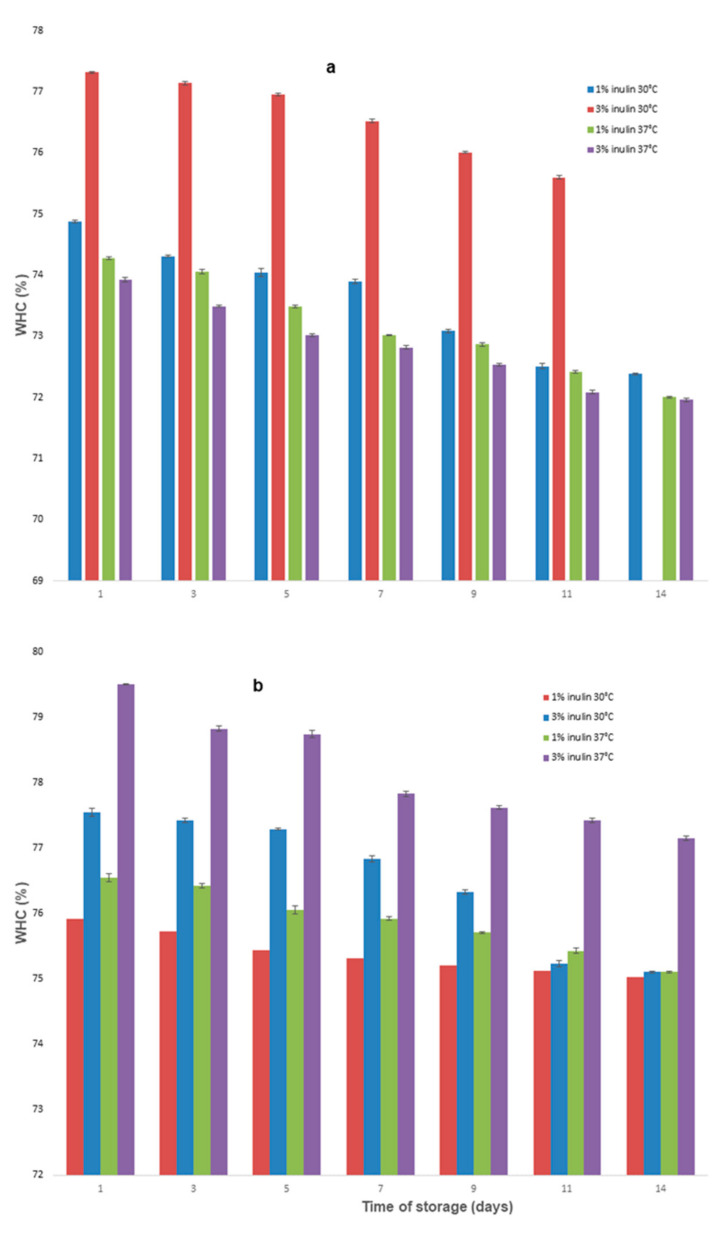
Variations in WHC of the beverages during storage. (**a**) Samples with SBS; (**b**) Samples with SBP.

**Table 1 microorganisms-11-00736-t001:** Physicochemical analysis of the samples before fermentation.

Sample	Soluble Solids (°Brix)	pH	Acidity (g Lactic Acid/100 mL)	Conductivity (µs/cm)
1% inulin and 20% SBS	4.25 ± 0.353 ^c^	4.21 ± 0.014 ^d^	0.44 ± 0.014 ^c^	5.92 ± 0.007 ^b^
3% inulin and 20% SBS	5.1 ± 0.141 ^a^	4.01 ± 0.014 ^a^	0.29 ± 0.028 ^a^	5.55 ± 0.014 ^a^
1% inulin and 3% SBP	4.1 ± 0.141 ^a^	4.11 ± 0.007 ^c^	0.35 ± 0.021 ^b^	6.05 ± 0.035 ^c^
3% inulin and 3% SBP	5.4 ± 0.283 ^b^	4.07 ± 0.021 ^b^	0.54 ± 0.028 ^d^	5.55 ± 0.014 ^a^

^a,b,c,d^ Different superscript letters in the same column indicate statistically significant difference (*p* < 0.05) according to Duncan’s multiple-range test. Values are expressed as means ± SD (n = 3).

**Table 2 microorganisms-11-00736-t002:** Rheological parameters of taste samples.

Sample	Viscosity (Pa s^−1^)	
	Fermentation at 30 °C	Fermentation at 37 °C
20% sea buckthorn syrup	0.39 ± 0.007 ^b^	0.42 ± 0.014 ^a^
1% sea buckthorn powder	1.74 ± 0.042 ^a^	1.97 ± 0.035 ^a^
3% sea buckthorn powder	1.86 ± 0.035 ^b^	2.11 ± 0.085 ^b^

^a,b^ Different superscript letters in the same column indicate statistically significant difference (*p* < 0.05) according to Duncan’s multiple-range test. Values are expressed as means ± SD (n = 3).

**Table 3 microorganisms-11-00736-t003:** Sensory evaluation parameters.

Characteristics	Beverage Sample							
	SBS at 30 °C		SBS at 37 °C		SBP at 30 °C		SBP at 37 °C	
	1% Inulin	3% Inulin	1% Inulin	3% Inulin	1% Inulin	3% Inulin	1% Inulin	3% Inulin
**Color**	3.8 ^c^	5.3 ^a,b^	5.6 ^a,b^	5.9 ^a^	3.87 ^c^	5.3 ^a,b^	5.6 ^a,b^	5.9 ^a,b^
**Flavor**	4.9 ^b^	5.3 ^a,b^	5.3 ^a,b^	5.8 ^a^	5.1 ^b^	5.4 ^a,b^	5.6 ^a,b^	5.8 ^a^
**Taste**	4.9 ^b^	5.4 ^a^	5.3 ^a,b^	5.4 ^a^	4.8 ^b^	5.3 ^a,b^	5.3 ^a,b^	5.6 ^b^
**Texture**	4.7 ^b^	5.3 ^a,b^	5.4 ^a^	5.5 ^a^	4.8 ^b^	5.3 ^a,b^	5.4 ^a^	5.5 ^a^
**Overall acceptability**	4.9 ^b^	5.2 ^b^	5.9 ^a^	6.1 ^a^	5.1 ^b^	5.4 ^b^	5.9 ^a^	6.3 ^a^

^a,b,c^ Different superscript letters in the same rows indicate statistically significant difference (*p* < 0.05) according to Duncan’s multiple-range test. Values are expressed as means ± SD (n = 3).

## Data Availability

Not applicable.

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
