# Peer review of "A Novel Symbiotic Beverage Based on Sea Buckthorn, Soy Milk and Inulin: Production, Characterization, Probiotic Viability, and Sensory Acceptance"

_microorganisms, 2023, doi:10.3390/microorganisms11030736_

Round 1

Reviewer 1 Report

The topic of the manuscript: „A novel symbiotic beverage based on sea buckthorn, soy milk and inulin: production, characterization, probiotic viability, and sensory acceptance falls within the thematic scope of the MICROORGANISMS.

 The aim of the study was to was to develop a novel symbiotic beverage based on lactic acid bacteria, soy milk, sea buckthorn syrup or powder enriched with inulin.

The main remarks concerns to:

1. “Abstract” – requires clarification (line 17),

2. “Introduction” - supplementing references to the literature (line 50),

3. “Materials and methods” - require supplementing the description of the methodology (subchapter 2.2.),

4. “Results”:

- no correlation between the way the results are presented in the Fig. 1 and their description in the text,

- in the description under Table 1 it is stated that the results are mean and standard errors (SE), while in section 2.7 it is stated that these will be standard deviations (SD) (similar situation in Table 2 and Table 3);

- necessary changes to Table 2 (no results for some of the tested variants) and Table 3;

- in the description under Table 3 it is stated that the statistical analysis concerns values in columns, while it concerns data in rows, unless the authors made a substantive error and the analysis is performed incorrectly at all;

5. “Discussion” - many fixes - incorrect citation,

6.     Among the performed determinations, the conductivity is mentioned, the results are presented in Table 1 but are not discussed at all in the text.

All suggestions for corrections (not only mentioned above) were introduced in the review mode to the attached pdf file.

Author Response

All the answers are in the attached document.

Reviewer 2 Report

The findings are of great interest for the industrials. The developed beverage can easily be transposed to an industrial scale. 

Here are some suggestions:

Line 22 - ssp. without Italic. But please check this in the entire manuscript.

Line 38 - ``From a scientific point of view, beverages are liquids produced for human consumption.`` - This sentence says nothing important for the topic described in the manuscript. 

Lines 36-42 - The introduction must state the importance of developing this novel beverage. The market of dairy-free beverages is only superficially described, and it is not directly referred to. Please refer also to similar types of beverages. You can cite some recent articles here and see the way the market is pointed out: 

Antonella Pasqualone, Carmine Summo, Barbara Laddomada, Elena Mudura, Teodora Emilia Coldea, Effect of processing variables on the physico-chemical characteristics and aroma of borÅŸ, a traditional beverage derived from wheat bran, Food Chemistry, Volume 265, 2018, Pages 242-252, ISSN 0308-8146, https://doi.org/10.1016/j.foodchem.2018.05.095

Keșa, A.-L.; Pop, C.R.; Mudura, E.; Salanță, L.C.; Pasqualone, A.; Dărab, C.; Burja-Udrea, C.; Zhao, H.; Coldea, T.E. Strategies to Improve the Potential Functionality of Fruit-Based Fermented Beverages. Plants 2021, 10, 2263. https://doi.org/10.3390/plants10112263

Claude P Champagne, Adriano Gomes da Cruz, Monica Daga, Strategies to improve the functionality of probiotics in supplements and foods, Current Opinion in Food Science, Volume 22, 2018, Pages 160-166, ISSN 2214-7993, https://doi.org/10.1016/j.cofs.2018.04.008

Line 43 - Carefully check the Latin denominations. Hippophae L. (L. not Italic)

Line 53 - ``soy drinks are the most consumed`` - prove it with number (percent/worldwide consumption)

Line 55 - not true. The cost (including production, marketing, etc.) is higher for dairy-free beverages. 

Line 66 - principal motive to be replaced with the main reason

Phrase 66-68 - has to be rephrase as it is unclear

Line 70 - not all fermented food could be functional as there are many types of fermentations. You have to be more clear here.

Line 72 - ``In the specialized literature`` - strange formulation in an article. It seems that this manuscript is not a specialized one. Please rephrase

Line 73 - ``alive`` replace with viable

Line 73 - `` sufficient numbers`` - incorrect formulation

Line 76- The use of the word ``additive`` is unclear here.

Lines 77-80 - Authors could do more so this part is clearer. The aim of the study is one of the most important parts of a paper. If it is unclear, the reader will not be interested in following the presented data in your paper. Please rephrase it.

Lines 88-91 - The soy beverage description is ambiguous. Please rephrase. What happens with the rest of the ingredients up to 100%?

Line 94 - What do you mean by ``untreated``?

Line 102 - Which is the producing country? Please prove with citing references that such long storage (2020 - 2022?. I believe this is the interval as we are already in 2023) of these raw materials did not affect their chemical composition. 

Line 105 - ssp. not Italic

The description of the fermentation process is incomplete and creates misunderstanding. Please reformulate. Possibly add a figure where to summarize all the obtaining processes, including the resulting samples. 

Line 133 - ssp. not Italic

The improvement of figures resolution is necessary.

The discussion section contains only a few comparisons to previously published papers. 

Author Response

All the modifications are in the attached file bellow
